# Plasma and Urine Levels of Glycosaminoglycans in Patients with Systemic Sclerosis and Their Relationship to Selected Interleukins and Marker of Early Kidney Injury

**DOI:** 10.3390/jcm11216354

**Published:** 2022-10-27

**Authors:** Kornelia Kuźnik-Trocha, Katarzyna Winsz-Szczotka, Katarzyna Komosińska-Vassev, Agnieszka Jura-Półtorak, Adrian Miara, Przemysław Kotyla, Krystyna Olczyk

**Affiliations:** 1Department of Clinical Chemistry and Laboratory Diagnostics, Faculty of Pharmaceutical Sciences in Sosnowiec, Medical University of Silesia, Jedności 8, 41-200 Sosnowiec, Poland; 2Department of Internal Medicine, Rheumatology and Clinical Immunology, Faculty of Medical Sciences in Katowice, Medical University of Silesia, 40-055 Katowice, Poland

**Keywords:** systemic sclerosis, glycosaminoglycans, chondroitin/dermatan sulfates, heparan sulfates/heparin, hyaluronic acid, keratan sulfates, interleukin 17, interleukin 18

## Abstract

Systemic sclerosis (SSc) is a chronic connective tissue disease characterized by immune system dysfunction, vasculopathy, and progressive fibrosis of the skin and internal organs, resulting from excessive accumulation of extracellular matrix (ECM) elements, including collagen and proteoglycans (PGs). An uncontrolled PG proliferation, caused by disturbances in their metabolism in tissues, is most likely reflected in the quantitative changes of their components, i.e., glycosaminoglycans (GAGs), in body fluids. Therefore, the aim of this study was to quantify the different types of GAGs in the blood and urine of systemic sclerosis patients. Chondroitin/dermatan sulfates (CS/DS) and heparan sulfates/heparin (HS/H) were quantified by hexuronic acid assay and electrophoretic fractionation, while hyaluronic acid (HA) and keratan sulfates were evaluated using ELISA tests. In turn, individual urinary GAGs were determined using the Blyscan™ Sulfated Glycosaminoglycan Assay Kit. Our results showed that the plasma concentrations of CS/DS, HS/H, HA, and KS in systemic sclerosis patients were significantly higher compared with those in healthy subjects. In the case of urine measurements, we have found that in SSc patients, CS/DC concentrations were significantly higher, while HA concentrations were significantly lower compared with the values observed in the urine of healthy subjects. Importantly, the found by us correlations between plasma keratan sulfate levels and both the duration of the disease and the severity of skin lesions, as expressed by the Rodnan scale, seems to suggest this GAG as a potential marker in assessing disease progression and activity. In addition, a level of urinary excretion of all types of GAGs due to their high positive correlation with uACR, may be a valuable complementary test in the diagnosis of early renal dysfunction in the course of SSc.

## 1. Introduction

Systemic sclerosis (SSc) is a multi-organ disease of chronic and progressive character that belongs to the group of systemic connective tissue diseases. The essence of this disease is progressive fibrosis of the skin and internal organs, mainly the heart, lungs, and kidneys [1,2]. The pathogenetic basis of systemic sclerosis is damage to capillary endothelium and immunological disorders, manifested by the synthesis of specific autoantibodies and a permanent inflammatory process [3,4]. Activated inflammatory cells excessively synthesize and secrete proinflammatory cytokines, including IL-1, IL-6, or TNF-α, which are capable of stimulating endothelial cells and fibroblasts. Thus, the chronic inflammatory process leads on the one hand to endothelial dysfunction and on the other, to the initiation of fibrosis [3,4,5,6]. The latter process is also influenced by interleukins released by activated T lymphocytes, i.e., IL-17 and IL-18, which show the ability to regulate it, exerting pro- and anti-fibrotic effects [3,4,6]. Although IL-17 and its downstream pathways have been shown to be strongly associated with the initiation and propagation of fibrosis, literature evidence suggests that this interleukin even though involved in fibrosis of most body organs in the course of SSc, does not exert an influence on the kidney [6]. However, in the latter case, there is evidence both for and against the role of the IL-17 pathway in directing fibrosis in this organ [7,8]. Additionally, the mechanism of action of IL-18 in the fibrosis process is controversial, as both pro- and antifibrotic effects have been described in the literature [2]. Thus, the exact role of the mentioned interleukins remains to be clarified. It is known, however, that fibrosis characteristic of SSc, resulting from cytokine stimulation of fibroblasts, involves excessive biosynthesis and formation of abnormal collagen deposits being resistant to tissue degradation. Fibrogenesis in SSc is believed to be caused not only by excessive collagen biosynthesis and accumulation but also by other components of the extracellular matrix (ECM), including proteoglycans (PGs) and their constituents, i.e., glycosaminoglycans (GAGs) [9,10]. The latter macromolecules, depending on the chemical structure of their chains, can be divided into the following types, i.e., chondroitin/dermatan sulfates (CS/DS), heparan sulfates/heparin (HS/H), keratan sulfates (KS) and hyaluronic acid (HA) [10,11]. Since the changes in PG/GAG metabolism observed in the course of systemic sclerosis, which depend on various factors, are probably reflected in the amounts of these compounds in body fluids, the aim of our study was to evaluate quantitatively particular fractions of GAGs, i.e., CS/DS, HS/H, KS, and HA, in blood plasma and urine of patients with systemic sclerosis. In addition, we decided to investigate the relationship between the concentrations of particular types of plasma glycosaminoglycans and the concentrations of IL-17 and IL-18, i.e., cytokines with a multidirectional environment-dependent mode of action that may play a potential role in the pathogenesis of this disease. In turn, we correlated the concentration of particular types of GAG in urine with microalbuminuria as an early marker of kidney damage. Moreover, taking into account GAGs as potential markers in the assessment of the severity of disease we investigated the relationship between the duration of the disease, the extent of skin lesions expressed by the Rodnan index (mRss), and the value of the Medsger scale, and the concentration of particular types of the aforementioned compounds in plasma and urine of patients with systemic sclerosis.

## 2. Materials and Methods

### 2.1. Characteristics of the Study Population

The study population consisted of 104 subjects, including 44 healthy controls (HC) and 60 patients with diffuse cutaneous systemic sclerosis who fulfilled the 2013 ACR/EULAR criteria for SSc [12], undergoing periodic follow-up at the Department of Internal Medicine, Rheumatology, and Clinical Immunology, the Medical University of Silesia in Katowice. The clinical characteristics of the groups are shown in Table 1.

Patients’ skin involvement was measured using the modified Rodnan skin score (mRss). The degree of skin thickness is measured in 17 body areas on a scale from 0 (normal) to 3 (severe), for a total score range of 0–51 [13]. Disease duration was calculated from the moment of the onset of the first clear clinical manifestation of SSc (excluding Raynaud’s phenomenon). Diagnostic procedures for specific organ and systemic involvement were performed. We evaluated the disease severity using clinical and laboratory parameters according to the Medsger Severity Scale. Inclusion criteria were the absence of renal involvement defined by a serum creatinine level < 1.3 mg/dL and normal urinary total protein excretion (<150 mg/day). In order to evaluate the degree of albuminuria to detect abnormalities at an earlier stage of the disease process in the kidneys, the albumin/creatinine ratio (uACR) was calculated. Microalbuminuria (MAU) was diagnosed in subjects whose uACR values ranged from 30 to 300 mg/g creatinine [14]. Based on the uACR results, the patients were divided into two groups, i.e., normoalbuminuric patients (SSc1) and microalbuminuric patients (SSc2). Patients underwent variable treatment regimens including the administration of immunosuppressants, according to EULAR/EUSTAR recommendations [15]. As appropriate, patients were given steroids not exceeding 10 mg/day and immunosuppressants in standard doses (for methotrexate < 25 mg/week; mycophenolate mofetil < 2.0 g/day; azathioprine < 200 mg/day). Patients currently taking biologics and nintedanib were excluded from the study. The exclusion criteria were a history of alcohol abuse, overlap syndromes, acute or chronic infections, diseases of the liver and kidney, cancer, and other systemic diseases. The study was conducted in accordance with the Declaration of Helsinki, and the Local Ethical Committee of the Silesian Medical University in Katowice approved the study protocol (KNW/0022/KB/208/14). All the patients provided written informed consent to participate in this study. No conflicts of interest have occurred during the implementation and completion of the study.

### 2.2. Preparation of Samples for Testing

A total of 4 mL of fasting venous blood was sampled from the subjects and placed in blood collection tubes containing sodium citrate anticoagulant (extraction and determination of plasma CS/DS and HS/H) and sodium heparin anticoagulant (measurement of plasma HA, KS, IL-17, and IL-18 levels). The blood was then centrifuged for 10 min at 1500× *g* at 4 °C. The plasma obtained in the above way was frozen and stored at −80 °C, until the start of the study.

Urine samples from the first-morning micturition were collected in 10 mL amounts into sterile containers and then centrifuged for 15 min at 15,000× *g*, at 25 °C, to separate morphotic elements, minerals and impurities. The resulting supernatant was stored at −80 °C until glycosaminoglycans were measured.

### 2.3. Analysis of CS/DS and HS/H in Plasma

The estimation of CS/DS and HS/H was performed using glycosaminoglycan samples isolated from plasma. A multistep extraction and purification process of GAGs including papain hydrolysis, β-elimination, and coupling with cetylpyridinium chloride was used. The quantitative assessment of glycosaminoglycans was performed by evaluating the concentration of hexuronic acids, co-forming the disaccharide chain sequences of these compounds, based on a calibration curve performed for standard solutions of D-glucuronolactone with concentrations of 1, 2, 4, 5, 10, 20 and 40 μg/mL. The isolated and purified GAG samples, in the concentration of 10 μg hexuronic acids, were then subjected to electrophoretic separation on cellulose acetate in 0.017 M aluminum sulfate solution, pH 2.6, using a current of 5 V and 1 mA. Electrophoretic separation of GAGs was performed both before and after the application of agents that specifically depolymerize specific types of these compounds, i.e., chondroitinase ABC, chondroitinase AC, and the combination of chondroitinase ABC with heparinase I and heparinase III, respectively. Electrophoretically separated individual GAG fractions, including CS/DS and HS/H, were quantified using the G:BOX Syngene Company (Frederick, MD, USA) gel documentation and analysis system. The detailed procedure was previously described [11,16].

### 2.4. The Assay of Urinary CD/DS and HS/H Concentration

Quantitative assessment of the concentration of sulfated glycosaminoglycans, i.e., CS/DS and HS/H in urine was performed using the Blyscan™ Sulfated Glycosaminoglycan Assay Kit (BioColor Ltd., Northern Ireland, UK), according to the manufacturer’s instructions. The principle of the assay is based on the spectrophotometric measurement of the absorbance of complexes consisting of GAGs, present in urine samples, and 1,9-dimethylmethylene blue (DMB), at λ = 656 nm. To determine the concentration of CS/DS, a procedure was performed to isolate these glycosaminoglycans from the total pool of GAGs in urine. This involved removing the HS/H fraction from the samples using nitric acid. Quantification of HS/H in urine was carried out using a mathematical formula. The total concentration of glycosaminoglycans in urine was determined with an analytical sensitivity of 0.5 mg/100 mL. The intra-assay coefficient of variation was less than 6.7%. Urine GAG concentrations were expressed in terms of g of creatinine (mg/g Cr).

### 2.5. Analysis of HA and KS 

HA levels in plasma and urine were measured in duplicate using a TECO^®^ Hyaluronic Acid PLUS ELISA kit (TECOmedical Group, Sissach, Switzerland), according to the product instructions. The analytical sensitivity was at 13.3 ng/mL. The appropriate low and high control samples were used for the quality control procedure. The intra-assay variation of the HA levels was less than 6%. Colorimetric detection was performed by a Tecan Infinite M200 Plate Reader (Tecan, Mannedorf, Switzerland). The results were analyzed with the help of Magellan 7.2 Software.

KS levels in plasma were measured in duplicate using Human KS ELISA Kit (Uscn Life Science Inc., Wuhan, China). The limit detection of the assay was 0.078 ng/mL, and the intra-assay coefficient of variation was < 7%. Due to the negligible amount of keratan sulfate in urine, this fraction of GAG was not tested in the urine of patients with systemic sclerosis.

### 2.6. Plasma IL-17 and IL-18 Analysis

The level of IL-17 in plasma was detected by enzyme-linked immunosorbent assay (ELISA), and the assessment was carried out with the reference to the instruction manual of a human IL-17 ELISA kit (Diaclone SAS, Besançon, France). The analytical sensitivity was at 2.3 pg/mL. The appropriate low and high control samples were used for the quality control procedure. The intra-assay variation of the IL-17 levels was less than 3.3%.

IL-18 plasma concentrations were determined by the sandwich ELISA method using the Human IL-18 ELISA Kit (eBioscience Corporation, Vienna, Austria). The analytical sensitivity was at 9 pg/mL. The appropriate low and high control samples were used for the quality control procedure. The intra-assay variation of the IL-18 levels was less than 8.1%.

### 2.7. Statistical Analysis

Statistical analysis was carried out using Statistica 13.0 package (StatSoft, Cracow, Poland). The normality of distribution was verified with the Shapiro–Wilk test. As the variables did not have a normal distribution, the results were expressed as medians and interquartile ranges. The nonparametric Mann–Whitney U test was used to assess differences between unrelated variables (control group and patients). Comparisons of more than two independent samples (control group and two patient groups) were performed using the nonparametric Kruskal–Wallis test and post hoc multiple comparison tests. Pearson’s correlation coefficient was employed for the statistical analysis of correlations between two variables. *p*-values of <0.05 were considered significant.

## 3. Results

### 3.1. The Plasma Levels of CS/DC, HS/H, HA, KS, IL-17, and IL-18 in Healthy Controls and Patients with SSc

The results of our study showed that the predominant types of GAGs in the plasma of both healthy and diseased subjects were chondroitin/dermatan sulfates. The plasma concentrations of all the investigated glycosaminoglycans, i.e., CS/DS, HS/H, HA, and KS, in patients with systemic sclerosis were significantly higher compared with healthy subjects, as shown in Table 2. Furthermore, we found that patients with normoalbuminuria showed higher plasma CS/DS, HS/H, and HA compared with patients with microalbuminuria (*p* < 0.05), whereas plasma KS levels were comparable between patients with SSc1 and SSc2 (Table 2.)

### 3.2. The Urinary Levels of CS/DC, HS/H, HA, KS, IL-17, and IL-18 in Healthy Controls and Patients with SSc

In contrast, particular types of GAGs in urine showed a different trend. While chondroitin/dermatan sulfate levels in the urine of SSc patients significantly increased (*p* < 0.00000) as in the blood, compared with healthy subjects, no significant differences were found between SSc1 and SSc2 patients. There were also no significant statistical differences between urinary heparan sulfates/heparin levels in systemic sclerosis patients and healthy subjects or between patients without microalbuminuria and those with MAU. However, there was a decrease in urinary hyaluronic acid excretion in systemic sclerosis patients compared with healthy subjects (*p* < 0.05), which was due to a decrease in HA concentration in the urine of SSc patients with microalbuminuria (*p* < 0.000000). In addition, we showed that urinary excretion of uHA was significantly higher in SSc patients with normoalbuminuria compared with patients with microalbuminuria (*p* < 0.05).

### 3.3. Correlation Analysis between Plasma CS/DS, HS/H, HA, KS and IL-17 and IL-18 in Patients with SSc 

Our study also reported that plasma IL-17 and IL-18 increased significantly in patients with SSc compared with healthy individuals. However, there were no differences in plasma IL-17 and IL-18 concentrations between SSc1 and SSc2 patients. In order to demonstrate the possible relationship between glycan components of the ECM, leading to the fibrosis characteristic of SSc, and cytokines, i.e., IL-17 and IL-18, that have been shown to regulate the fibrosis process, a statistical correlation analysis was performed, the results of which are presented in Table 3.

As can be seen from the obtained results (Table 3), an average negative correlation between CS/DS and IL-17 concentrations in the blood of patients with SSc (r = −0.45; *p* < 0.05) was demonstrated. This relationship was also present in patients without microalbuminuria (r = −0.52; *p* < 0.05) as well as in those with MAU (r = −0.40; *p* < 0.05), as shown in Figure 1a.

The correlation between IL-17 and KS levels in the plasma of SSc patients was different as compared with that concerning CS/DS. There was a positive average correlation between the examined parameters (r = 0.34; *p* < 0.05) in all SSc patients (Table 3) and in SSc2 patients (r = 0.34; *p* < 0.05) (Figure 1b). There was no correlation between the levels of mentioned interleukin, HS/H, and HA in blood. However, although there was no correlation between IL-17 and HS/H levels in all SSc patients (Table 3), opposite correlations were found in patients with and without microalbuminuria (Figure 1c). Thus, in SSc patients with normoalbuminuria a positive average correlation between IL-17 concentration and HS/H concentration was found (r = 0.43; *p* < 0.05), whereas in SSc patients with microalbuminuria a negative correlation between the parameters studied was observed (r = −0.40; *p* < 0.05).

The assessment of the relationship between CS/DS, HS/H, HA and KS, and IL-18 in the plasma of patients with SSc, showed no correlation between the analyzed parameters (*p* > 0.05), which is presented in Table 3. Only a positive correlations were found between plasma IL-18 and hyaluronic acid (r = 0.51; *p* < 0.05) and keratan sulfates (r = 0.49; *p* < 0.05) levels in SSc patients with microalbuminuria (Figure 1d,e).

### 3.4. Correlation Analysis between Urinary (u) CS/DS, HS/H, HA, and uACR in Patients with SSc

On the other hand, correlation analysis between particular types of GAGs in urine and albumin/creatinine ratio revealed positive correlations between the parameters studied. Thus, a positive, high correlation was found between uCS/DS and uACR (r = 0.63; *p* < 0.05) in all SSc patients (Table 3), which was even higher in patients with microalbuminuria (r = 0.91; *p* < 0.05) (Figure 2a). Thus, a positive, high correlation was found between uCS/DS and uACR (r = 0.63; *p* < 0.05) in all SSc patients (Table 3), which resulted only from very strong relationship between variables in patients with microalbuminuria (r = 0.91; *p* < 0.05) (Figure 2a). A similar relationship was confirmed between uHS/H and uACR (r = 0.62; *p* < 0.05) in SSc patients (Table 3), which showed a value slightly lower in SSc1 patients (r = 0.52; *p* < 0.05) and higher in SSc2 patients (r = 0.71; *p* < 0.05), as illustrated in Figure 2b. In addition, there was a positive high correlation between uHA and uACR in SSc patients with MAU (r = 0.80; *p* < 0.05) (Figure 2c).

### 3.5. Correlation Analysis between Plasma and Urine Concentrations of Particular Types of Glycosa-Minoglycans and Disease Duration, Rodnan Skin Score and Medsger Scale in Patients with SSc

In order to assess the suitability of GAGs as potential useful markers in the assessment of disease activity, the relationship between the duration of the disease and the extent of skin lesions, as expressed by the Rodnan index (mRss), and the concentrations of particular types of GAGs in the plasma and urine of patients with systemic sclerosis were examined. The obtained results are presented in Table 4.

As shown in Table 4, plasma chondroitin/dermatan sulfate concentration displayed only a negative high correlation with mRss value (r = −0.51; *p* < 0.05) in SSc patients with normoalbuminuria. The same patients also showed a positive average correlation between plasma keratan sulfate concentration and skin lesion severity (r = 0.44, *p* < 0.05). A similar correlation was found between plasma hyaluronic acid concentration and modified Rodnan skin score (r = 0.33, *p* < 0.05) but solely in SSc patients with microalbuminuria.

When the relationship between plasma concentrations of individual types of GAGs and disease duration was evaluated, a negative weak correlation in all SSc patients (r = −0.34; *p* < 0.05) was found only for keratan sulfates. The demonstrated relationship was present both without and with microalbuminuria (Table 4).

## 4. Discussion

Systemic sclerosis is a connective tissue disease of a chronic and progressive character, the essence of which is fibrosis of the skin and internal organs, initiated by inflammation, vascular changes, and immunological disorders. As a result, patients experience significant hardening and thickening of the skin and damage to internal organs [1,2,3,4,5,6]. As mentioned earlier, ECM components, including proteoglycans and their components, i.e., glycosaminoglycans, play an important role in fibrosis. In the course of the disease, there is an increase in PG/GAG deposition in the skin, the body organ most subject to fibrogenesis [17,18,19,20]. Previous studies evaluating GAGs in skin taken from patients with systemic sclerosis have also shown the existence of abnormal proportions of the different types of glycosaminoglycans compared with the skin of healthy subjects [18,19,20]. Thus, Fleischmajer et al. [17] showed that the increase in the total amount of GAGs in the skin of patients with SSc was the result of increased content of only dermatan sulfate, the main skin glycosaminoglycan besides hyaluronic acid. Similar results were obtained by Akimoto et al. [18], Higuchi et al. [19], and Yokoyama et al. [20], who further demonstrated that increased DS compactness was accompanied by a reduction in the amount of hyaluronic acid in the skin of SSc patients. In contrast, the opposite trend of changes in HA content in the skin of SSc patients was demonstrated by Søndergaard et al. [21] and Juhlin et al. [22]. In addition, Uitto et al. [23] found that the increased content of sulfated glycosaminoglycans in the skin of SSc patients was due to increased chondroitin sulfate synthesis. It was also shown that an increase in the content of chondroitin/dermatan sulfates in the skin of SSc patients correlated with the severity of skin lesions [19]. Moreover, quantitative evaluation of the skin heparan sulfates of patients with systemic sclerosis showed a slight increase in the content of the aforementioned compounds, but only in tissue samples obtained from areas without visible fibrotic lesions [24]. These results suggest that the quantitative changes in HS/H probably represent an expression of early disorders in the remodeling of the heteropolysaccharide components of the connective tissue ECM during the course of the disease. The demonstrated quantitative changes in the different types of GAGs occurring in the skin, and probably in other tissues and organs of systemic sclerosis patients, seem to reflect both enhanced biosynthesis and impaired degradation of the mentioned macromolecules. The tissue (cellular and extracellular) metabolic turnover of PGs in the course of systemic sclerosis is most likely reflected by quantitative changes in their components, i.e., glycosaminoglycans in the blood. According to our previous studies, patients with SSc experience a significant plasma increase in total GAG concentration compared with healthy subjects [10]. In these studies, we showed that systemic sclerosis patients have increased blood concentrations of all types of these compounds-including chondroitin/dermatan sulfates, heparan/heparin sulfates, keratan sulfates, and hyaluronic acid. However, as reported in the literature, the only GAG fraction whose quantitative changes in the blood of systemic sclerosis patients have been previously studied was hyaluronic acid [25]. Although most researchers, like us, have shown an increase in this glycosaminoglycan in the blood of patients with SSc [21,26,27,28] Neudecker et al. [25] found that the increase in serum HA concentrations applies only to patients in the early stages of SSc, i.e., with a disease duration of up to 2 years. They also showed that serum hyaluronidase (Hyal-1) activity in SSc patients decreases with disease duration [25]. Hyal-1, found in parenchymal organs, fibroblasts, and keratinocytes of the skin, as well as blood and urine, is responsible for the hydrolysis of hyaluronic acid [25]. These data could explain the gradual tissue HA accumulation progressing along with a disease duration; however, the no relationship between HA concentration and disease duration demonstrated in this study does not seem to support this thesis. Nevertheless, the hypothesis put forward by Engström-Laurent et al. [26], according to which hyaluronic acid synthesis in tissues may be stimulated by PDGF, which is released in excessive amounts from platelets during the course of the disease, seems plausible. The same authors did not show a correlation between increasing serum HA concentration and the extent of skin lesions accompanying the disease, which is partially consistent with our study. We found a weak correlation between plasma HA and mRss value only in the group of patients with microalbuminuria, not in all the patients. Moreover, in the same patients, we also demonstrated the existence of a positive correlation between the plasma concentration of hyaluronic acid and IL-18. The mentioned cytokine is probably involved in the pathogenesis of the disease, as indicated by the elevated IL-18 values shown in the blood of patients with SSc both by us and other authors [3,4]. In addition, Kitasato et al. [29] found that IL-18 mediates liver fibrosis through the activation of CD4+ T cells and that this effect is blocked by anti-IL-18 treatment. On the other hand, it was observed that in renal fibrosis, proximal tubular cells stimulated by IL-18 can induce the α-SMA, collagen I, and fibronectin production in a dose- and time-dependent manner [30]. Thus, it could not be excluded that this interleukin also shows the ability to stimulate hyaluronic acid synthesis by proximal tubular cells. Although the role of IL-18 in the development of scleroderma renal crisis (SRC), being a major complication in SSc patients, has not yet been confirmed, previous studies suggest that it may play an important role in fibrosis-induced renal damage. Indeed, it was found in one study that patients with high serum IL-18 levels had significantly higher serum creatinine concentrations [3]. However, since renal impairment can be present in SSc patients despite normal serum creatinine concentration, the search for new markers of renal damage seems particularly important [31]. Interestingly, in SSc patients with MAU, plasma levels of IL-18 were also positively correlated with elevated plasma keratan sulfate levels. Such correlations were not found in all patients, and in patients with normoalbuminuria, which may suggest the use of these GAGs as a biochemical marker of renal changes, however further large-sample-size studies are needed to confirm this hypothesis.

Notwithstanding, the results of our study, concerning changes in KS, CS/DS and HS/H concentrations in the blood of SSc patients cannot be compared with the other results because the above-mentioned ECM macromolecules have not been evaluated so far. It is worthy of note that there are reports on changes in glycosaminoglycan concentration in the blood of patients with another systemic autoimmune connective tissue disease, i.e., rheumatoid arthritis (RA) [16]. These studies, like ours, showed an increase in plasma concentrations of all types of GAGs in patients with high disease activity [16]. In both SSc and RA, autoimmune disorders exacerbated by pro-inflammatory cytokines (IL-1, IL-6, TNF-α), growth factors (TGF-β), and reactive oxygen species (ROS) play an important role in the development of the mentioned diseases [10,16]. The aforementioned cytokines and growth factors, stimulating GAG biosynthesis, may lead to increases in plasma CS/DS and HA levels in SSc patients. The IL-6 and TGF-β seem to rather stimulate CS/DS synthesis, while IL-1 and TNF-α are involved especially in the up-regulation of a HA expression by the nuclear factor-kB pathway [11]. The found by us increase in the concentration of HS/H in the blood of normoalbuminuric SSc patients, which positively correlated with the concentration of IL-17, may also suggest the involvement of this cytokine in the induction of heparan sulfate synthesis at an early stage of the disease. It cannot be ruled out that the increase in plasma concentrations of individual GAGs in subjects with systemic sclerosis is also a manifestation of increased non-enzymatic degradation of tissue PGs, stimulated by reactive oxygen species. Oxidative stress has been found in SSc patients [32]. It is noteworthy, that the increase in the concentration of CS/DS and HS/H in the blood of microalbuminuric SSc patients, was negatively correlated with the concentration of interleukin 17. The latter is a cytokine that has been found to be involved in the development of fibrosis in the course of SSc in most body organs except the kidneys [6]. Therefore, the demonstrated negative association of IL-17 with CS/DS and HS/H in the blood of MAU patients may indirectly suggest a role for this interleukin pathway in directing changes associated with renal fibrosis.

Taking into account that GAGs are eliminated from circulation by liver endothelial cells and the renal route, the increased plasma pool of individual GAGs of SSc patients should be accompanied by the increased urinary excretion of the mentioned glycan compounds, particularly CS/DS as the most abundant fraction of plasma GAGs [10,25]. Our results seem to be in line with that assumption as we have found that the concentration of chondroitin-dermatan sulfates in the urine of SSc patients from all the study groups is significantly increased compared with healthy subjects. The increase in urinary CS/DS excretion in SSc patients observed in this study may be related to abnormalities in the PG/GAG metabolism in the renal capillary walls. It is known that the disease is accompanied by endothelial dysfunction and fibrosis of the renal vessel structures resulting from the deposition of type I, III, and IV collagen as well as fibronectin in the inner capillary layer [33,34,35,36]. In contrast, the observed by us a significant reduction in hyaluronic acid concentration in the urine of SSc patients compared with healthy subjects may be due to impaired HA metabolism in the affected kidneys. We showed that HA concentration in the urine of MAU patients was significantly lower compared with both healthy subjects and SSc patients without microalbuminuria. The reduced urinary excretion of HA may be due, on the one hand, to a decreased plasma hyaluronidase activity progressing along with the duration of the disease [25], and, on the other hand, to an increase in the course of SSc production of reactive oxygen species (ROS) [32,37]. It cannot be excluded that an increased, ROS-stimulated degradation of hyaluronic acid, leading to a fragmentation of this glycan and subsequent accumulation of the resulting HA fragments in the kidneys, promotes inflammation in the organ in question, exacerbating its damage and impairing the urinary excretion of this compound in SSc patients [38,39]. It should be mentioned that a sensitive indicator of early renal damage in SSc patients is an uACR index based on an albuminuria assessment [2]. The high positive correlations shown in our study between the uACR value and the urinary concentration of all evaluated types of GAGs in microabuminuric SSc patients may be a valuable complementary test in the diagnosis of early renal dysfunction and in monitoring the effectiveness of the applied treatment in the course of SSc.

## 5. Conclusions

Our study provides information on the quantitative changes in the different types of glycosaminoglycans in the blood and urine of patients with systemic sclerosis. Moreover, the demonstrated relationship between plasma keratan sulfate levels and both disease duration and skin lesion severity, as expressed by the Rodnan skin score, suggests the possibility of using this GAG as a potential marker in assessing disease progression and activity. 

## Figures and Tables

**Figure 1 jcm-11-06354-f001:**
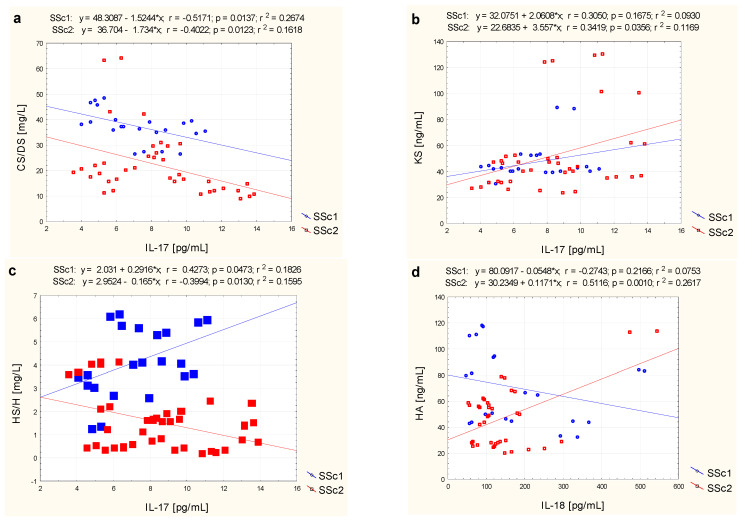
Correlation analysis between IL-17 and sequentially CS/DS (**a**), HS/H (**b**), and KS (**c**) and between IL-18 and sequentially HA (**d**) and KS (**e**) in the blood of patients with systemic sclerosis and normoalbuminuria (SSc1) or microalbuminuria (SSc2) presented as a scatter plot with fitted regression line.

**Figure 2 jcm-11-06354-f002:**
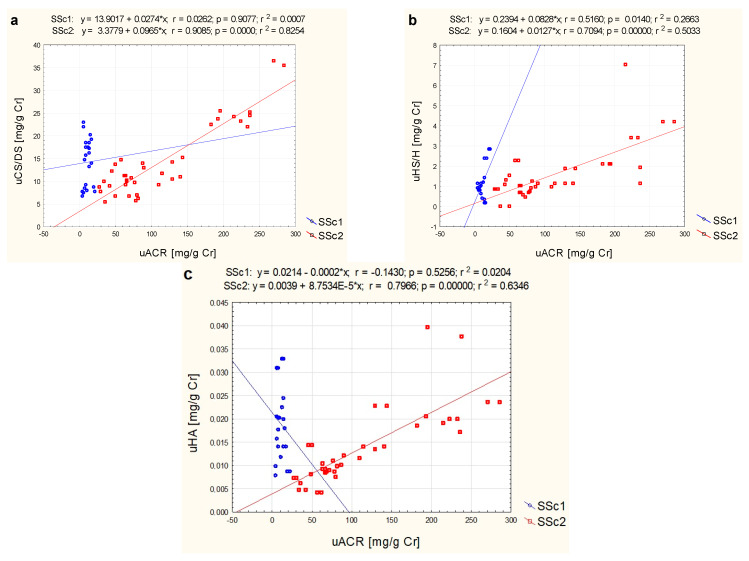
Correlation analysis between urinary (u) CS/DS (**a**), HS/H (**b**), HA (**c**), and uACR in patients with systemic sclerosis and normoalbuminuria (SSc1) or microalbuminuria (SSc2), presented as a scatter plot with fitted regression line.

**Table 1 jcm-11-06354-t001:** Demographic, clinical and treatment characteristics of systemic sclerosis patients and healthy controls.

Parameter	HC (n = 44)	SSc (n = 60)
Age (years)	54.40 ± 5.06 *	52.85 ± 14.29 *
Gender (F:M)	34:10	45:15
BMI (kg/m^2^)	24.87 ± 3.74	23.93 ± 4.42
Disease duration (years)	NA	2.5 (1.00, 6.00) ^●^
Rodnan skin score (mRss)	NA	25.00 (11.00, 33.00) ^●^
Medsger scale	NA	6.00 (4.00, 8.00) ^●^
Laboratory parameters		
WBC (10^3^/μL)	7.24 ± 2.32 *	7.48 ± 3.00 *
RBC (10^6^/μL)	4.86 ± 0.51 *	4.31 ± 0.42 *
PLT (10^3^/μL)	228.78 ± 76.16 *	215.73 ± 87.91 *
Hb (g/dL)	15.46 ± 2.19 *	12.55 ± 1.44 *
Ht (%)	42.68 ± 4.53 *	37.59 ± 3.92 *
Total protein (g/dL)	6.18 ± 0.94 *	6.35 ± 0.82 *
Glucose (mg/dL)	88.60 ± 6.10 *	86.29 ± 12.87 *
γ-globulins (g/L)	9.58 ± 2.76 *	17.99 ± 5.67 *
Cr (mg/dL)	0.74 ± 0.09 *	0.78 ± 0.13 *
CRP (mg/L)	0.78 (0.40, 1.40) ^●^	5.00 (4.00, 9.00) ^●^
ESR (mm/h)	15 (12.00, 20.00) ^●^	18 (13.00, 29.00) ^●^
ANA	100% (negative)	100% (positive)
Scl-70	100% (negative)	57% (positive)
uAlb (mg/dL)	1.12 (0.80–2.00) ^●^	11.97 (1.54–20.23) ^●^
uCr (g/dL)	0.16 (0.13–0.17) ^●^	0.18 (0.11–0.30) ^●^
uACR (mg/g Cr)	7.84 (6.02–11.53) ^●^	51.18 (13.53–80.56) ^●^
Clinical characteristics		
Arterial hypertension, n (%)	NA	52 (86.67)
Interstitial lung disease (ILD), n (%)	NA	44 (73.33)
Digital ulceration, n (%)	NA	33 (55.00)
Esophagus dysmotility, n (%)	NA	20 (33.33)
Arthralgia, n (%)	NA	17 (28.33)
Arrhythmia, n (%)	NA	11 (18.33)
Pulmonary arterial hypertension (PAH), n (%)	NA	7 (11.67)
Treatment		
Mycophenolate mofetil (MMF), n (%)	NA	35 (58.33)
Cyclophosphamide (CYC), n (%)	NA	24 (40.00)
Metotrexate (MTX), n (%)	NA	12 (20.00)
Azathioprine (AZA), n (%)	NA	7 (11.67)
Bosentan, n (%)	NA	0 (0%)

*—Results are expressed as mean±SD; ^●^—results are expressed as medians (quartile 1–quartile 3); HC, healthy controls; NA, not applicable, WBC, white blood cell; RBC, red blood cell; Hb, hemoglobin; Ht, hematocrit; PLT, platelet; Cr, creatinine; CRP, C-reactive protein; ESR, erythrocyte sedimentation rate; ANA, antinuclear antibodies; Scl-70, antibodies against topoisomerase I; uAlb, urinary albumin; uCr, urinary creatinine; uACR, urinary albumin/creatinine ratio.

**Table 2 jcm-11-06354-t002:** The distribution patterns of plasma CS/DS, HS/H, HA, KS, IL-17, IL-18 and urinary (u) CS/DS, HS/H, HA in the healthy controls and SSc patients.

Parameter	Healthy Controls HC (n = 44)	SSc Patients		
		All Patients SSc (n = 60)	Patients without Microalbuminuria SSc1 (n = 22)	Patients with Microalbuminuria SSc2 (n = 38)
CS/DS (hexuronic acids, mg/L)	9.60 (6.96–11.49)	26.49 (16.66–37.32) ^a^	37.32 (34.90–39.46) ^a^	18.51 (13.03–25.76) ^b,f^
HS/H (hexuronic acids, mg/L)	0.83 (0.36–1.42)	2.07 (0.80–4.03) ^c^	4.06 (3.14–5.62) ^a^	1.47 (0.51–2.12) ^g^
HA (ng/mL)	22.93 (17.37–27.81)	50.60 (29.42–67.66) ^a^	65.68 (44.65–93.75) ^a^	48.67 (27.62–58.33) ^a,h^
KS (ng/mL)	27.80 (23.60–31.30)	42.45 (36.50–52.10) ^a^	42.25 (40.60–52.60) ^a^	42.90 (32.20–51.60) ^a^
IL-17 (pg/mL)	5.99 (1.44–7.56)	7.85 (5.60–9.75) ^d^	7.20 (5.29–8.82)	8.23 (5.64–10.84) ^d^
IL-18 (pg/mL)	92.39 (76.47–129.87)	117.90 (86.53–177.67) ^e^	119.48 (73.54–293.74) ^e^	116.61 (91.44–164.51) ^e^
uCS/DS (mg/g Cr)	3.26 (2.78–3.70)	12.61 (8.88–18.48) ^a^	15.21 (8.48–18.44) ^a^	11.21 (9.22–22.08) ^a^
uHS/H (mg/g Cr)	1.25 (0.59–1.87)	1.07 (0.79–1.89)	0.96 (0.63–1.22)	1.13 (0.88–2.12)
uHA (mg/g Cr)	0.022 (0.010–0.029)	0.014 (0.009–0.020) ^e^	0.019 (0.014–0.022)	0.011 (0.008–0.019) ^a,h^
uACR (mg/g Cr)	7.84 (6.02–11.53)	53.63 (13.60–112.18) ^a^	9.88 (6.49–14.01)	80.87 (62.43–182.07) ^a,i^

All results are expressed as medians (quartile 1-quartile 3); ^a^ *p* < 0.000000, ^b^ *p* < 0.000005, ^c^ *p* < 0.00005, ^d^ *p* < 0.005, ^e^ *p* < 0.05 compared with control group; ^f^ *p* < 0.005, ^g^ *p* < 0.00005, ^h^ *p* < 0.05, ^i^ *p* < 0.000000 compared with SSc1 patients; SSc1, systemic sclerosis patients without microalbuminuria; SSc2, patients with microalbuminuria; CS/DS, chondroitin/dermatan sulfate; HS/H, heparan sulfate/heparin; HA, hyaluronic acid; KS, keratan sulfates; IL-17, interleukin 17; IL-18, interleukin 18; uCS/DS, urinary chondroitin/dermatan sulfate; uHS/H, urinary heparan sulfate/ heparin; uHA, urinary hyaluronic acid; uACR, urinary albumin/creatinine ratio.

**Table 3 jcm-11-06354-t003:** Correlation analysis between plasma CS/DS, HS/H, HA, KS and IL-17 and IL-18 as well as urinary (u) CS/DS, HS/H, HA and uACR in patients with systemic sclerosis (Pearson’s correlation coefficients, r).

Plasma		
Parameter	IL-17	IL-18
	r (*p*)	r (*p*)
CS/DS	−0.451 (0.0003)	0.168 (NS)
HS/H	−0.223 (NS)	0.071 (NS)
HA	−0.176 (NS)	0.176 (NS)
KS	0.343 (0.007)	0.203 (NS)
	Urine	
	uACR	
	r (*p*)	
uCS/DS	0.632 (0.000000)	
uHS/H	0.618 (0.000000)	
uHA	0.282 (0.029)	

NS, not significant (*p* ≥ 0.05); CS/DS, chondroitin/dermatan sulfate; HS/H, heparan sulfate/heparin; HA, hyaluronic acid; KS, keratan sulfates; IL-17, interleukin 17; IL-18, interleukin 18; uCS/DS, urinary chondroitin/dermatan sulfate; uHS/H, urinary heparan sulfate/heparin; uHA, urinary hyaluronic acid; uACR, urinary albumin/creatinine ratio.

**Table 4 jcm-11-06354-t004:** Correlation analysis between plasma and urine concentrations of particular types of glycosaminoglycans in patients with systemic sclerosis and disease duration and Rodnan skin score (Pearson correlation coefficients, r).

Parameter	Duration of Disease		Modified Rodnan Skin Score		Medsger Scale	
	SSc		SSc		SSc	
	r (*p*)		r (*p*)		r (*p*)	
CS/DS	0.073 (NS)		0.002 (NS)		−0.082 (NS)	
HS/H	0.026 (NS)		0.105 (NS)		0.116 (NS)	
HA	−0.091 (NS)		0.233 (NS)		−0.097 (NS)	
KS	−0.336 (0.009)		0.022 (NS)		−0.112 (NS)	
uCS/DS	0.001 (NS)		0.025 (NS)		−0.104 (NS)	
uHS/H	−0.063 (NS)		0.066 (NS)		−0.156 (NS)	
uHA	−0.038 (NS)		−0.023 (NS)		0.028 (NS)	
	SSc1	SSc2	SSc1	SSc2	SSc1	SSc2
	r (*p*)	r (*p*)	r (*p*)	r (*p*)	r (*p*)	r (*p*)
CS/DS	0.297 (NS)	−0.093 (NS)	−0.510 (0.015)	0.195 (NS)	0.123 (NS)	−0.241 (NS)
HS/H	−0.205 (NS)	0.032 (NS)	0.193 (NS)	0.199 (NS)	0.008 (NS)	0.171 (NS)
HA	−0.110 (NS)	−0.190 (NS)	0.210 (NS)	0.328 (0.045)	−0.283 (NS)	−0.014 (NS)
KS	−0.335 (0.031)	−0.366 (0.024)	0.442 (0.039)	-0.097 (NS)	−0.354 (NS)	−0.040 (NS)
uCS/DS	0.335 (NS)	−0.154 (NS)	0.076 (NS)	0.005 (NS)	−0.262 (NS)	−0.039 (NS)
uHS/H	0.053 (NS)	−0.074 (NS)	−0.271 (NS)	0.149 (NS)	−0.133 (NS)	−0.161 (NS)
uHA	0.162 (NS)	−0.236 (NS)	0.109 (NS)	−0.071 (NS)	0.013 (NS)	0.007 (NS)

NS, not significant (*p* < 0.05); SSc1, systemic sclerosis patients without microalbuminuria; SSc2, patients with microalbuminuria; CS/DS, chondroitin/dermatan sulfate; HS/H, heparan sulfate/heparin; HA, hyaluronic acid; KS, keratan sulfates; IL-17, interleukin 17; IL-18, interleukin 18; uCS/DS, urinary chondroitin/dermatan sulfate; uHS/H, urinary heparan sulfate/heparin; uHA, urinary hyaluronic acid.

## Data Availability

The datasets analyzed or generated during the study are available from the corresponding author.

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
