# Peer review of "Plasma and Urine Levels of Glycosaminoglycans in Patients with Systemic Sclerosis and Their Relationship to Selected Interleukins and Marker of Early Kidney Injury"

_jcm, 2022, doi:10.3390/jcm11216354_

Round 1

Reviewer 1 Report

In this manuscript, Kornelia Kuźnik-Trocha and colleagues report the results of their study investigating the relationship between glycosaminoglycans and other markers of inflammation and fibrosis in 60 patients with SSc and 44 healthy subjects. They observed higher plasma concentrations of CS/DS, HS/H, HA and KS in SSc patients. Moreover, plasma keratan sulfate levels correlated with both the duration of the disease and the severity of skin lesions.

There are a few points that I would like to raise.

1. Study population. I understand that all of your patients have the diffuse form of SSc. How were patients selected? Were there any patients treated with MMF, Rituximab, tocilizumab, bosentan and so on?

2. Table 1 should be improved. Complete demographic, clinical and treatment characteristics of the patients should be reported.

3. It is not clear why the Authors choose to investigate only IL-17 and IL-18 and why they divided the patients according to the uACR. Please, explain.

4. Results. Please divide results in sub-paragraphs to make them more readable.

5. Figures. The figures appear to be the result of a regression plot rather a Pearson correlation (a scatter plot that should have no line). Please, correct.

English needs moderate revision by a native speaker.

Author Response

Answers to the comments of the Reviewer 1

We would like to thank for the Reviewer’s valuable and detailed evaluation of our paper entitled: “Plasma and urine levels of glycosaminoglycans in patients with systemic sclerosis and their relationship to selected interleukins and marker of early kidney injury” by K. Kuźnik-Trocha et al.

The Reviewer comment 1:

Study population. I understand that all of your patients have the diffuse form of SSc. How were patients selected? Were there any patients treated with MMF, Rituximab, tocilizumab, bosentan and so on?

Answer to the Reviewer comment 1:

We have added specific inclusion and exclusion criteria to the manuscript and also provided more detailed information about the treatment of all the patients treatment:

  1. Line 100-102: “Inclusion criteria were the absence of renal involvement defined by a serum creatinine level < 1.3 mg/dl and normal urinary total protein excretion (<150 mg/day).”
  2. Line: 111-112: “Patients currently taking biologics and nintedanib were excluded from the study.”
  3. Line 109-111: “As appropriate, patients were given steroids not exceeding 10 mg/day and immunosuppressants in standard doses (for methotrexate < 25 mg/week; mycophenolate mofetil < 2.0 g/day; azathioprine < 200 mg/day).”

The Reviewer comment 2:

Table 1 should be improved. Complete demographic, clinical and treatment characteristics of the patients should be reported.

Answer to the Reviewer comment 2:

In accordance with the Reviewer's comments, Table1 was supplemented with clinical characteristics and treatment information.

The Reviewer comment 3:

It is not clear why the Authors choose to investigate only IL-17 and IL-18 and why they divided the patients according to the uACR. Please, explain.

Answer to the Reviewer comment 3:

IL-17 and IL-18 were chosen for the study due to the fact that the relationship between these interleukins and glycosaminoglycans has not been described so far.

Details have been included in the Introduction section. Line 47-56 and 69-73.

In addition, SSc patients qualified according to the inclusion criteria without renal lesions were divided into two groups according to the uACR, in order to assess whether abnormalities in early-stage renal disease could affect blood and urine glycosaminoglycan levels.

The below presented sentences have been introduced into the text of the manuscript.

Line 100-104: “Inclusion criteria were the absence of renal involvement defined by a serum creatinine level< 1.3 mg/dl and normal urinary total protein excretion (<150 mg/day). In order to evaluate the degree of albuminuria to detect abnormalities at an earlier stage of the disease process in the kidneys, the albumin/creatinine ratio (uACR) was calculated.”

The Reviewer comment 4:

Results. Please divide results in sub-paragraphs to make them more readable.

Answer to the Reviewer comment 4:

According to the Reviewer suggestion, the results have been divided into sub-paragraphs.

The Reviewer comment 5:

Figures. The figures appear to be the result of a regression plot rather a Pearson correlation (a scatter plot that should have no line). Please, correct.

Answer to the Reviewer comment 5:

As suggested by the Reviewer, the figure titles have been changed as presented below:

  1. Line 251-254: “Figure 1. Correlation analysis between IL-17 and sequentially CS/DS (a), HS/H (b) and KS (c) and between IL-18 and sequentially HA (d) and KS (e) in the blood of patients with systemic sclerosis and normoalbuminuria (SSc1) or microalbuminuria (SSc2) presented as a scatter plot with fitted regression line.”
  2. Line 286-288: “Figure 2. Correlation analysis between urinary (u) CS/DS (a), HS/H (b), HA (c) and uACR in patients with systemic sclerosis and normoalbuminuria (SSc1) or microalbuminuria (SSc2), presented as a scatter plot with fitted regression line.”

The Reviewer comment 6:

English needs moderate revision by a native speaker.

Answer to the Reviewer comment 6:

English native speaker has reviewed manuscript and language mistakes have been corrected and modified.

Reviewer 2 Report

An interesting study on the levels and relationship of glycosaminoglycans and cytokines IL-17 and IL-18 in patients with SSc.

It is interesting to note that 100% of the included patients had positive ANA and Scl-70 antibodies. If we know that antiScl-70 antibodies are positive in 20-40% of patients with SSc, whether positive ANA and Scl-70 antibodies were included criteria of the study.

In Tables 2, 3 and 4 there is no explanation of the abbreviations used. They should be added

Interpretations and explanations of the results in the discussion are mostly speculative (hypothetical). They need to be simplified.

1. KS levels in the group of patients with MAU and without MAU do not differ significantly, nor do the levels of IL 17 and IL 18. Therefore, it is difficult to conclude, based only on the described correlations, that the mentioned GAG could be used as an indicator of kidney damage. It is necessary to correct part of the text from lines 356 to 367 on page 10.

2. If reduced synthesis of heparan proteoglycan (HSPG) leads to increased permeability of the basement membrane of glomerular capillaries and increased urine filtration, how to explain the absence of a significant difference in CS/DS levels in patients with MAU and without MAU. Explain in the text or leave out the sentences in lines 405-409 on page 11.

In conclusion, I would leave out may be a valuable complementary test in the diagnosis of early renal dysfunction in SSc. Despite the significant correlation with uACR, the levels of the mentioned GAGs do not differ significantly in patients with MAU and without MAU.

Author Response

Answers to the comments of the Reviewer 2

We would like to thank for the Reviewer’s valuable and detailed evaluation of our paper entitled: “Plasma and urine levels of glycosaminoglycans in patients with systemic sclerosis and their relationship to selected interleukins and marker of early kidney injury” by K. Kuźnik-Trocha et al.

The Reviewer comment 1:

An interesting study on the levels and relationship of glycosaminoglycans and cytokines IL-17 and IL-18 in patients with SSc.

Answer to the Reviewer comment 1:

Thank you, we appreciate the encouraging words.

The Reviewer comment 2:

It is interesting to note that 100% of the included patients had positive ANA and Scl-70 antibodies. If we know that antiScl-70 antibodies are positive in 20-40% of patients with SSc, whether positive ANA and Scl-70 antibodies were included criteria of the study.

Answer to the Reviewer comment 2:

Thank you very much for your valuable comment, as it allowed us to put up the correct number of SSc patients with anti-Scl-70 antibodies into the table. Unfortunately, when editing Table 1 in the manuscript, a value of 100% was mistakenly entered into the column presenting Scl-70 antibodies, while it should be: 57% (as in that percent of patients we detected the presence of these antibodies). We are very sorry for this mistake.

Furthermore, we would like to kindly inform that detailed inclusion criteria have been implemented into the text of  Materials and Methods section of the manuscript in sub-paragraph entitled: Characteristics of the Study Population. The information concerning the inclusion criteria is as follows:  " Inclusion criteria were the absence of renal involvement defined by a serum creatinine level < 1.3 mg/dl and normal urinary total protein excretion (<150 mg/day).”

The Reviewer comment 3:

In Tables 2, 3 and 4 there is no explanation of the abbreviations used. They should be added

Answer to the Reviewer comment 3:

In accordance with the Reviewer's comment, the explanations of the abbreviations used have been added under the Tables 2, 3 and 4.

The Reviewer comment 4:

Interpretations and explanations of the results in the discussion are mostly speculative (hypothetical). They need to be simplified.

  1. KS levels in the group of patients with MAU and without MAU do not differ significantly, nor do the levels of IL 17 and IL 18. Therefore, it is difficult to conclude, based only on the described correlations, that the mentioned GAG could be used as an indicator of kidney damage. It is necessary to correct part of the text from lines 356 to 367 on page 10.

Answer to the Reviewer comment 4.1:

As suggested by the Reviewer, the above fragment of the text has been changed into:

“Interestingly, in SSc patients with MAU, plasma levels of IL-18 were also positively correlated with elevated plasma keratan sulfate levels. Such correlations were not found in all patients, and in patients with normoalbuminuria, which may suggest the use of these GAGs as a biochemical marker of renal changes, however further large sample size studies are needed to confirm this hypothesis.”

  1. If reduced synthesis of heparan proteoglycan (HSPG) leads to increased permeability of the basement membrane of glomerular capillaries and increased urine filtration, how to explain the absence of a significant difference in CS/DS levels in patients with MAU and without MAU. Explain in the text or leave out the sentences in lines 405-409 on page 11.

Answer to the Reviewer comment 4.2:

According to the Reviewer’s suggestion, the sentences in question (lines 405-409 on page 11) have been left out.

The Reviewer comment 5:

In conclusion, I would leave out may be a valuable complementary test in the diagnosis of early renal dysfunction in SSc. Despite the significant correlation with uACR, the levels of the mentioned GAGs do not differ significantly in patients with MAU and without MAU.

Answer to the Reviewer comment 5:

In accordance with the Reviewer's suggestions, the indicated sentence has been left out.

Reviewer 3 Report

This paper reports interesting findings. I would love to see the correlation of these results with the presence or absence of clinical findings characteristic of SSc, such as interstitial pneumonia, gastroesophageal reflux disease, and the presence or absence of ulcers.

Author Response

Answers to the comments of the Reviewer 3

We would like to thank for the Reviewer’s valuable and detailed evaluation of our paper entitled: “Plasma and urine levels of glycosaminoglycans in patients with systemic sclerosis and their relationship to selected interleukins and marker of early kidney injury” by K. Kuźnik-Trocha et al.

The Reviewer comment 1:

This paper reports interesting findings. I would love to see the correlation of these results with the presence or absence of clinical findings characteristic of SSc, such as interstitial pneumonia, gastroesophageal reflux disease, and the presence or absence of ulcers.

Answer to the Reviewer comment 1:

We made an attempt to examine the relationship between  blood and urine glycosaminoglycan levels in patients with SSc and the Medsger scale (which was added to Table 4), but no correlation was found between these parameters.

Round 2

Reviewer 1 Report

The Authors have addressed the issues raised in the previous report. I have no further comments.